# Genome-Wide Identification and Expression Analysis Unveil the Involvement of the Cold Shock Protein (CSP) Gene Family in Cotton Hypothermia Stress

**DOI:** 10.3390/plants13050643

**Published:** 2024-02-26

**Authors:** Yejun Yang, Ting Zhou, Jianglin Xu, Yongqiang Wang, Yuanchun Pu, Yunfang Qu, Guoqing Sun

**Affiliations:** 1College of Agronomy, Shanxi Agricultural University, Jinzhong 030800, China; 15903469823@163.com (Y.Y.); zz15034151584@163.com (T.Z.); 2Biotechnology Research Institute, Chinese Academy of Agricultural Sciences, Beijing 100081, China; 15334197038@163.com (J.X.); wcy694687@163.com (Y.W.); 3College of Agronomy, Xinjiang Agricultural University, Urumqi 830052, China; 4Institute of Western Agriculture, The Chinese Academy of Agricultural Sciences, Changji 831100, China; puyuanchun1209@126.com

**Keywords:** cotton, cold shock domain protein, abiotic stress, cold acclimation, expression analysis

## Abstract

Cold shock proteins (CSPs) are DNA/RNA binding proteins with crucial regulatory roles in plant growth, development, and stress responses. In this study, we employed bioinformatics tools to identify and analyze the physicochemical properties, conserved domains, gene structure, phylogenetic relationships, cis-acting elements, subcellular localization, and expression patterns of the cotton *CSP* gene family. A total of 62 CSP proteins were identified across four cotton varieties (*Gossypium arboreum*, *Gossypium raimondii*, *Gossypium barbadense*, *Gossypium hirsutum*) and five plant varieties (*Arabidopsis thaliana*, *Brassica chinensis*, *Camellia sinensis*, *Triticum aestivum*, and *Oryza sativa*). Phylogenetic analysis categorized cotton CSP proteins into three evolutionary branches, revealing similar gene structures and motif distributions within each branch. Analysis of gene structural domains highlighted the conserved CSD and CCHC domains across all cotton *CSP* families. Subcellular localization predictions indicated predominant nuclear localization for *CSPs*. Examination of cis-elements in gene promoters revealed a variety of elements responsive to growth, development, light response, hormones, and abiotic stresses, suggesting the potential regulation of the cotton *CSP* family by different hormones and their involvement in diverse stress responses. RT-qPCR results suggested that *GhCSP.A1*, *GhCSP.A2*, *GhCSP.A3*, and *GhCSP.A7* may play roles in cotton’s response to low-temperature stress. In conclusion, our findings underscore the significant role of the *CSP* gene family in cotton’s response to low-temperature stress, providing a foundational basis for further investigations into the functional aspects and molecular mechanisms of cotton’s response to low temperatures.

## 1. Introduction

Low temperatures are a significant environmental limitation on production and geographic distribution globally for plants, impacting growth and development [1]. Tropical and subtropical plants are more damaged by cold temperatures in general compared to overwintering plants with high cold tolerance. Low-temperature stress limits plant gene expression through direct inhibition of metabolic responses and indirect osmotic, oxidative, and other stresses [2]. To avert adverse effects, over the course of evolution, plants have devised sophisticated mechanisms to resist cold stress through the integration of transcription factors [3,4,5].

Contemporary research has demonstrated that freezing is tolerated by plants following cold exposure for an extended period, a process known as cold acclimation [6]. Cold domestication affects ion concentrations and metabolite transport, initiating downstream cold signaling [7]. The ICE-CBF-COR pathway is the most extensively studied regulatory pathway for low-temperature tolerance in plants today. The Inducer of CBF Expression (ICE) is a key factor in cold acclimation, and the regulatory gene C-repeat binding factors (CBF) plays a crucial role in cold acclimation. ICE1 induces the expression of CBF3 at low temperatures, while CBF can activate the expression of COR genes by binding to the CRT element (TGGCCCGAC) in the cold-responsive (COR) promoter [8]. Cold-inducible genes improve cold tolerance in plants by regulating the synthesis of compatible solutes (soluble sugars and proline), pigments (lutein and carotenoids), and cold-responsive proteins such as antifreeze proteins (AFPs), late embryogenesis abundance (LEAs) proteins, heat shock proteins (HSPs), and cold shock proteins (CSPs) [9,10,11,12].

Cold shock proteins (CSPs) act as DNA/RNA binding proteins that are conserved across bacterial and eukaryotic domains, including higher plants [13,14,15]. All plant *CSPs* are structurally similarly, with the cold shock domain (CSD) located at the N-terminus, and a large glycine-rich region at the C-terminus, which is randomly interspersed with 2–7 zinc finger domains (CCHC-type) [1,16]. CSP proteins have a role in cell proliferation, stress adaption, transcription and translation control, and cell division [17]. CSP proteins facilitate translation and destabilize RNA secondary structures to function as RNA chaperones, aiding the body in adapting to low-temperature stress [18].In eukaryotes, cold shock structural domain proteins are pleiotropic, and plant *CSPs*, in addition to being regulated by cold stress, are implicated in the regulation of embryo development, flowering time, fruit development, and stress responses, and this may be a function of the auxiliary structural domains in eukaryotic CSP proteins in addition to CSDs [16]. 

Up to now, the *CSP* family has only been systematically analyzed in *Arabidopsis thaliana*, *Brassica chinensis*, *Camellia sinensis*, *Triticum aestivum*, and *Oryza sativa*. In these species, four, six, three, two, and four *CSP* genes have been identified, respectively [18,19,20,21,22].The first *CSP* gene was identified in winter *T. aestivum*, and this *WCSP1* gene was upregulated following low-temperature induction, possibly involved in regulating the cold tolerance of winter wheat. However, it was not regulated by other environmental stresses, suggesting that the function of *WCSP1* was specific to cold adaptation [1]. Two cold shock structural domain proteins in *O. sativa* were cloned and functionally characterized, and it was determined that the expression of *OsCSP1* and *OsCSP2* was most elevated in rice spike tissues, as well as flowers and seeds. The expression of *OsCSP1* and *OsCSP2* was upregulated in rice root and stem tissues within 24 h of low-temperature stress treatment [23]. Overexpression of *AtCSP3* enhances the tolerance and survival of transgenic plants exposed to multiple stresses, including salt, drought, and low temperatures [24,25]. Given that many *CSP1*-associated mRNAs are implicated in ribosome biogenesis, CSP1 may be essential for growth at low temperatures above freezing, with overexpression of *AtCSP1* enhancing the translation of its targets under water limitation stress [26]. Overall, the *CSP* family proteins participates in the regulation of plant stress response and growth and development processes.

Cotton is a globally cultivated cash crop, with both diploid and allotetraploid varieties. *Gossypium hirsutum* and *Gossypium barbadense* are allotetraploids with the genetic composition AADD, which originated from the A genome of the diploid cotton *Gossypium arboreum* and the D genome of *Gossypium raimondii* [27]. Low-temperature cold damage is a common abiotic stress in cotton growing areas, where seed germination and seedling growth and development are inhibited, seriously impacting crop yield and quality. In cotton-producing areas of Xinjiang, the agrometeorological event with the greatest impact on cotton yield and quality is chilling, which occurs during the seedling stage in the Spring [28]. To improve responses to low-temperature cold damage, enhancing cotton genetic germplasm resources and identifying cold-tolerant genes are necessary approaches. The expression of *GhDREB1* is positively regulated through the typical mechanisms of cold stress signaling, resulting in transgenic tobacco exhibiting higher cold tolerance compared to the wild type [29]. Overexpression of *GhRaf19* positively regulates cold tolerance by modulating reactive oxygen species in cotton [30]. While some progress has been made in the mining of cold tolerance genes in cotton, there are a limited number of effective genes identified to respond to low-temperature stress and applied in the field [31]. Therefore, it is critical to locate cotton cold tolerance genes, characterize the cold tolerance mechanism, and develop cotton germplasm resources for resistance to low-temperature stress.

To date, functional examination of cotton *CSP* genes in hormone signaling, growth, development, and stress response has not been documented. In this study, we examined both the structure and evolutionary patterns of the *CSPs* gene family according to genome-wide data from the four cotton species. We used the expression of the *CSP* family under stress to characterize their responses. The findings elucidate the functions of the *CSP* gene family in cotton, and lay the foundation for the identification of genes for improving low-temperature tolerance in cotton.

## 2. Results

### 2.1. Identification of CSP Family Members across Four Cotton Species

To characterize members of *CSPs* across four cotton species (*G. arboreum*, *G. raimondii*, *G. barbadense*, and *G. hirsutum*), a search was performed using the Hidden Markov Model (HMM) and CSP protein sequences from *Arabidopsis thaliana*. A total of 43 genes containing *CSP* domains were identified, and 8 genes from *G. boreum*, 8 genes from *G. aimondii*, 13 genes from *G. hirsutum*, and 14 genes from *G. barbadense* were found. Seven genes were identified in the A genome of *G. hirsutum* tetraploid and the A and D genomes of *G. borbadense*, respectively. In comparison, six genes were identified in the D genome of *G. hirsutum.* Tetraploid cotton is an allotetraploid formed by hybridizing diploids with At and Dt genomes [32].The number of *CSP* genes in tetraploid cotton (*G. hirsutum*, *G. barbadense*) was nearly twice that of diploid cotton (*G. arboreum*, *G. raimondii*), indicating that the *CSP* family is conserved throughout evolution. These results indicate that the number of *CSPs* varies slightly across different cotton species, potentially due to the loss or duplication of genes over the course of cotton evolution.

Based on the position of these genes in the cotton chromosome, they were named *GaCSP.1-8* (*G. arboreum*), *GrCSP.1-8* (*G. raimondii*), *GhCSP.A1-7* and *GhCSP.D1-6* (*G. hirsutum*), and *GbCSP.A1-7* and *GbCSP.D1-7* (*G. barbadense*). Appendix A indicates the gene ID number, coding region (CDS) length (bp), protein sequence length (aa), protein molecular weight (MW), isoelectric point (pI), and subcellular localization prediction results of the identified cotton *CSP* family proteins. The proteins encoded by the 43 genes had different physicochemical properties, varying in length from 90 to 300 amino acids, molecular weight from 8524.11 to 27,434.94 kDa, theoretical isoelectric points (pl) from 5.89 to 9.23, and subcellular localization predicted the localization of cotton CSP proteins primarily in the nucleus.

### 2.2. Analysis of Subcellular Localization of Cotton CSP Gene

*GhCSP.A1/A2/A3* was used as an example to validate subcellular localization. The vector was transiently expressed in tobacco leaves, and the *GhCSP.A1/A2/A3* proteins were all localized to the nucleus as determined by the GFP fluorescence signals generated by laser confocal microscopy, in agreement with the prediction (Figure 1). This is similar to the subcellular localization results of *A. thaliana AtCSP1~3* [25,26,33].

### 2.3. Gene Structure, Conserved Domains, and Motif Analysis of Cotton CSP Genes

Gene structure is essential for determining the phylogenetic relationships among *CSP* genes [34]. To examine the evolutionary relationships throughout *CSP* genes in *G. arboreum*, *G. raimondii*, *G. barbadense*, and *G. hirsutum*, a phylogenetic tree was constructed using 43 characterized CSP protein sequences. Our findings showed three groups of *CSP* genes across the four cotton species closely distributed in each group.

Gene structure, conserved structural domains, and motifs were closely linked to gene function [35]. Consequently, cotton *CSPs* were analyzed according to phylogenetic trees. The coding sequences were mapped to the exon/intron sequences of the cotton genome to determine the gene structure of the cotton *CSP* family. The analysis demonstrated that the exon/intron structure of the cotton *CSP* genes was conserved across varieties. To further study the phylogenetic relationship of *CSP* genes, we investigated the arrangement of exons and introns in the *CSP* gene in cotton (Figure 2C). All group II genes had only one exon, while some members of groups I and III genes had three or five exon-intron structures. The sequence structure examination indicated that the majority of cotton *CSP* genes were composed of one exon, while a few of them consisted of two to five exons and introns, which was consistent with previous studies [16]. The exon and intron structures of cotton *CSP* genes are highly similar, supporting the reliability of the phylogenetic analysis. To further understand the structural characteristics of cotton *CSP* genes, we characterized nine conserved motifs using MEME. After analysis, it was determined that motifs 1, 2, 3, 4, 6, 7, and 8 appeared in nearly all CSP protein sequences from the three groups. Motif 5 was unique to Group I, and Motif 9 was absent in Group II (Figure 2A). Groups I was identical except for *GrCSP.5*, which lacks motifs 4 and 9. The group II *CSP* gene contains precisely the same motifs, but some of the motifs were arranged differently, lacking motifs 5 and 9 compared to group I. The motifs of group III were essentially the same as those of group II, but some of the motifs in group III were different, including *GaCSP.4* and *GrCSP.4*, containing the lowest number of motifs. This finding demonstrates that members of the same group may differ in function. From the findings of conserved structural domains, similar to the *A. thaliana* CSP protein structural domains, all cotton CSP proteins contained both typical CSD and varied amounts of CCHC (Figure 2B). 

### 2.4. Chromosomal Location, Gene Duplication, and Evolutionary Relationships of Gossypium CSP Family Genes

Cotton *CSP* genes are not distributed across all 13 chromosomes. It was found that 13 *GhCSP* genes were located on 10 chromosomes, of which seven were located on five chromosomes of the At genome, and the remaining six were located on five chromosomes of the Dt genome, according to the annotation information in the cotton genome (Figure 3). *GhCSP.A1/D1*, *GhCSP.A2/D2*, and *GhCSP.A3* were located on chromosome A/D01, *GhCSP.A5/D3*, *GhCSP.A6/D5*, and *GhCSP.A7/D6* were located on chromosome A05, 11, and 12, respectively, while *GhCSP.A4* was located on chromosome A03 and *GhCSP.D4* was located on chromosome A06. It was determined that 14 *GbCSP* genes were located on 12 chromosomes, with seven genes in each of the At and Dt genomes. The distribution of these two groups of genes on chromosomes was essentially identical. *GbCSP.A1/D1* and *GbCSP.A2/D2* were located on the A/D01 chromosomes, *GbCSP.A4/D4*, *GbCSP.A5/D5*, *GbCSP.A6/D6*, and *GbCSP.A7/D7* were located on A/D05,6,11,12 chromosomes, respectively. Additionally, *GbCSP.A3* was located on chromosome A03, while *GbCSP.D3* was located on chromosome D02.

The *GaCSP* gene was localized on seven chromosomes, distributed across chromosomes 01 (*GaCSP.1*), 02 (*GaCSP.2* and *GaCSP.3*), 03 (*GaCSP.4*), 05 (*GaCSP.5*), 06 (*GaCSP.6*), 11 (*GaCSP.7)*, and 12 (*GaCSP.8*). The seven genes of *GrCSP* were situated on six chromosomes, including 02 (*GrCSP.1*, *GrCSP.2*, and *GrCSP.3*), 05 (*GrCSP.4*), 07 (*GrCSP.5*), 08 (*GrCSP.6*), 09 (*GrCSP.7*), and 10 (*GrCSP.8*). We determined that the A and D genomes of tetraploid cotton differed from diploid cotton in terms of the number of genes and chromosomal distribution, and these findings indicate that *CSP* genes may have undergone gene translocations, loss and duplication, and chromosomal deletions throughout the evolution of cotton.

To gain insight into the linkages between *CSP* genes and putative gene duplications in cotton genomes, we determined the non-synonymous (Ka) and synonymous (Ks) levels of 273 repeat gene pairs across 10 combinations of four cotton species. Selective pressure was predicted based on Ka/Ks values. Ka/Ks = 1 indicated neutral selection (pseudogenes), Ka/Ks < 1 indicated purification or negative selection, meaning that the gene evolution tends towards purification, and Ka/Ks > 1 indicated positive selection. The examination demonstrated that gene pairs with Ka/Ks > 1 appeared only in one pair throughout the Gb-Gb group, six gene pairs had Ka/Ks values between 0.5 and 1, while 233 gene pairs had Ka/Ks values below 0.5. Consequently, we hypothesized that the *CSP* gene family in cotton experienced strong purification selection (Appendix A).

To investigate the genetic origin and evolutionary relationship of *CSPs* throughout cotton, we performed intraspecific and interspecific collinear analyses. Intraspecific collinear analysis of the four cotton varieties indicated that the collinearity of diploid cotton was effectively identical the same, with the exception of *GrCSP.4* in *G. raimondii* being homologous to *GrCSP.5* but divergent in *G. arboreum*. The genetic collinearity of tetraploid cotton was effectively equal, but alterations throughout cotton evolution and chromosome doubling were observed (Figure 4A–D). For instance, *GhCSP.A1* does not have a homolog in the D genome, suggesting that it may have been ectopic or lost throughout evolution. Interestingly, *Ghir_A06G004400.1* was determined to have collinearity with *GhCSP.A5*, *GhCSP.D3*, and *GhCSP.D4*, but it did not belong to the *CSPs* family, potentially due to an incomplete domain following gene replication in the evolutionary process. We conducted interspecific comparisons with *A. thaliana* and *B. chinensis* using *G. hirsutum*. It was identified that upland cotton possessed seven homologous gene pairs with *A. thaliana* and eight homologous gene pairs with *B. chinensis*, suggesting that these genes have elevated structural and functional similarity (Figure 4E,F).

### 2.5. Phylogenetic Analysis

To investigate the evolutionary relationships of Cotton CSP proteins and those of five other plants, we created a neighbor-joining tree using multiple comparisons of *CSP* amino acid sequences. The phylogenetic tree indicated that *CSPs* had three distinct evolutionary branches, classified as CSP I–CSP III (Figure 5). CSP I contained 15 members, including two *GrCSPs* and *GaCSPs*, four *GbCSPs* and *GhCSPs*, two *CsCSPs*, and one *OsCSP*. CSP II possessed 19 members (three *WCSPs*, one *OsCSP*, two *AtCSPs*, two *BcCSPs*, three *GhCSPs*, four *GbCSPs*, two *GrCSPs*, and *GaCSPs*). CSP III was composed of the largest number of *CSP* members, including two *CsCSPs*, two *AtCSPs*, four *BcCSPs*, six *GbCSPs* and *GhCSPs*, and four *GrCSPs* and *GaCSPs*, for a total of 28. Prior reports have demonstrated that *CsCSP4* is upregulated at low temperatures, and *GhCSP.A7* is closely linked to *CsCSP4*, indicating that *GhCSP.A7* may have comparable functions [22]. A close relationship between *GhCSP.A4* and *AtCSP2* was documented, while *AtCSP2* has a negative regulatory role in low-temperature stress, *GhCSP.A4* may also play a negative regulatory role [36]. Remarkably, each branch in the *CSP* gene family exhibits homologous genes across all four cotton species, suggesting a close relationship among their *CSP* genes. Notably, the proportion of *CSPs* in allotetraploid *G. hirsutum* and *G. barbadense* nearly maintains a 1:1 ratio, mirroring the similar ratio observed in diploid *G. raimondii* and *G. arboreum*. The proportion of *CSPs* in allotetraploid cotton and diploid cotton was below 2:1, potentially because of evolutionary selection during the hybridization of two diploid cotton lines to generate allotetraploid cotton.

### 2.6. Cis-Element Analysis in the Promoter Regions of Cotton CSP Genes

Gene expression is often regulated by cis-elements in the upstream promoter sequences [37]. These cis-elements in the non-coding DNA upstream of the transcription start site can modulate stress receptors or tissue-specific expression of the gene in diverse environments [35]. To comprehend the potential regulatory mechanisms of this family, we predicted the cis-acting elements of 43 cotton CSP proteins and examined their upstream 2000 bp promoter regions using PlantCARE. The findings indicated that in addition to common essential core elements (such as the A-box and TATA-box) and light-responsive elements (such as ACE), the promoter region of *CSPs* had numerous elements related to growth, development, and abiotic stress response (Figure 6).

Hormone response elements include those that react to auxin (AuxRE), ABA (ABRE), ethylene (ERE), methyl jasmonate (CGGTA motif and TGACG motif), gibberellin (GARE-motif, P-box, and TATC-box), and salicylic acid (SARE and TCA). The stress response elements include low-temperature response elements (LTR), dehydration response elements (DRE), drought response elements (MBS, Myb/Myc binding sites), defense and stress response elements (TC-rich repeats), anaerobic induction elements (ARE), and wound response elements (WUN and WRE3 motif). Growth and developmental elements include a meristematic tissue expression element (CAT-box), an AT-rich element (binding site of AT-rich DNA-binding protein), and an endosperm expression element (GCN4 motif). The findings indicate that *CSPs* may be regulated by diverse hormones involved in the growth of cotton and its response to environmental stresses. Various subfamilies have different gene promoter characteristics, including those of subfamilies II and III, typically containing a large number of hormone and stress-responsive elements, while the promoters of subfamilies I and III contain a large number of growth and developmental elements. This indicates that the expression of *CSPs* in different subfamilies is influenced by diverse regulatory factors.

### 2.7. Analysis of GhCSP Gene Expression under Various Stresses and Different Tissues

Given that gene expression is closely related to cis-acting elements, we examined abiotic stress and tissue-specific expression patterns for *GhCSPs.* As there is only information about upland cotton found in the database, we characterized upland cotton. The gene expression heatmap indicated that the same *GhCSPs* were expressed under different stress treatments, while the expression patterns of homologous genes were relatively similar across different tissues (roots, stems, leaves, and floral organs) and under various stress conditions (salt, drought, low temperature, and high temperature) (Figure 7). Moreover, while genes from the same subfamily possess similar motifs, they may differ functionally under different abiotic stresses related to stress response elements in the gene promoter region.

### 2.8. Validation of GhCSPs Expression Levels

To investigate the tissue-specific expression of the *GhCSPs* gene family and its response to low-temperature stress, RT-qPCR verification and significance analysis were conducted on *GhCSPs*. Given the homology correlation, *GhCSP.A1-A7* was selected. From the RT-qPCR results, the tissue-specific expression results were nearly consistent with the RNA-seq data, and most of the *GhCSP* genes were highly expressed in the roots or stems, while *GhCSP.A5* and *GhCSP.A7* were specifically highly expressed in the torus (Figure 8). We simulated low-temperature stress, and the results indicated that the expression levels of *GhCSP.A1*, *GhCSP.A2*, *A3*, and *A7* were upregulated under low-temperature induction, except *GhCSP.A6* which had no significant change (Figure 9). The expression of *GhCSP.A1* was gradually upregulated following low-temperature treatment, reaching its highest level after 48 h, while the expression of *GhCSP.A2* reached its highest level 6 h after low-temperature treatment and then gradually decreased. *GhCSP.A3* expression was downregulated at 1 h and 6 h and significantly upregulated at 24 h, while *GhCSP.A7* expression was significantly upregulated at 12 h and 24 h. The expression of *GhCSP.A4* was downregulated, and *GhCSP.A5* was upregulated at 1 h and 6 h. In contrast, both were downregulated at 12 h and 24 h, with the highest expression taking place at 1 h and gradually decreasing. Based on these results, most genes in the cotton *CSP* family can respond to low-temperature stress via expression regulation.

## 3. Discussion

Low-temperature cold damage typically occurs throughout cotton growth and development, posing a significant threat and loss to agricultural production. Over the past two decades, efforts have been made to determine the pivotal elements of cold tolerance in plants, as well as analyze their regulatory mechanisms. Mining of low-temperature tolerance candidate genes has been important for this process [38]. A large number of studies have indicated that CSP proteins are ubiquitous in plants and play an essential role in adapting to various abiotic stresses (especially low-temperature stress) and regulating plant growth and development [16,23,39,40]. It was found that the number of *CSPs* in the four cotton varieties was nearly four times (*G. barbadense* and *G. hirsutum*) or twice (*G. arboreum* and *G. raimondii*) the number of *CSPs* identified in *A. thaliana*, *C. sinensis*, and *T. aestivum*. This difference may be attributed to the extensive gene amplification that cotton underwent to adapt to its environment during the evolutionary process.

The connection in the distribution of gene structures and conserved motifs in the same subfamily suggests that they have common roles in plant growth [41]. Upon studying the cotton CSP protein sequences, we discovered that the majority of *CSP* genes in cotton do not contain introns. Intronless genes are typically found in bacteria and primitive eukaryotes [35,42]. This suggests that *CSP* genes without introns are evolutionarily conserved and have been passed down from the past to the present. Furthermore, the CSD and CHCC zinc finger structural domains of cotton *CSPs* are highly conserved. These two domains can facilitate the binding of cold shock domain proteins to RNA, single-stranded DNA (ssDNA), and double-stranded DNA (dsDNA), demonstrating that cold shock domain proteins exhibit RNA chaperone activity [22].However, the motif distribution of cotton CSP proteins varies across subfamilies, and some motifs are specific to a particular subfamily. For instance, motif 5 is specific to subfamily I, which suggests that the functions of the genes may differ among subfamilies. 

Chromosome mapping and collinearity analysis of cotton *CSPs* demonstrated that cotton had undergone gene deletion and chromosome translocation during evolution and chromosome doubling. *Ghir_A06G004400.1* was found to be collinear with the *CSP* gene of upland cotton, but it was not identified as belonging to the *CSP* family. By examining the conserved domain, it lacks the CSD, the most critical cold shock domain in the *CSP* family, which may be lost during evolution. In addition, through interspecific comparison, we found that there were homologous gene pairs between cotton and dicotyledonous plants (*A. thaliana*, and *B. chinensis*), but not between cotton and monocotyledonous plants (*O. sativa*, and *T. aestivum*), related to species development.

Cis-acting elements on gene promoters can play an active role in plant stress responses, and the type of element dictates the response signal of the gene [43]. In the promoter sequences of the cotton *CSP* family, we identified various cis-acting elements in response to environmental stress (ARE, DRE, MBS, LTR, and WRE3, etc.), plant hormone response (ABRE, AuxRE, ERE, SARE, CGGTA motif, and P-box, etc.), and growth and development (GCN4 motif, and CAT-box, etc.). A large number of Myb and Myc transcription factor binding sites are present in the promoters of most *CSP* genes, and it has been demonstrated that these transcription factors are critical for plant stress tolerance [44,45].The distribution of these cis-acting elements on the promoter indicates that *CSPs* are likely to respond to adversity, stress, and growth and development.

Contemporary studies have indicated that *CSP* activity in plants is influenced by various abiotic stress factors, and the examination of gene expression patterns is often employed to characterize gene function [32]. Via analysis of expression patterns before and after stress, it was determined that the expression of four genes (*GhCSP.A1*, *GhCSP.A2*, *GhCSP.A3*, and *GhCSP.A7*) was upregulated under low temperatures. Similarly, the expressions of *BcCSP1* and *BcCSP3* in cabbage and *CsCSP1* and *CsCSP4* in Camellia were upregulated by low-temperature stress [20,22]. The expression of *GhCSP.A1* was significantly upregulated by cold stress, but downregulated by salt and drought stress. The expression of *GhCSP.A5* was upregulated under salt and drought stress, but downregulated after 6 h of low-temperature stress. Regrettably, no pleiotropic genes were identified in *GhCSPs*. Our laboratory acquired the *DgCSP* gene from Deinococcus gobiensis I-0, and subsequent overexpression in cotton confirmed its ability to increase yield and enhance drought and salt tolerance [46]. Studies have determined that following *AtCSP2* deletion, the cold tolerance of the *atcsp2* mutant did not exhibit any significant alterations. Still, overexpression of this gene reduced the cold resistance of transgenic *A. thaliana.* This negative regulatory mechanism is accomplished through inhibiting the expression of CBF (C-repeat binding factor) and its downstream genes [36]. Similar to *AtCSP2*, *GhCSP.A4* is also inhibited by low-temperature treatment, suggesting that this gene may negatively regulate cotton cold resistance response, but its involvement in CBF regulation and downstream genes remains unclear. These results further validate the significant role of the *GhCSP* gene family in responding to stress. Under low-temperature conditions, the upregulated expression of *GhCSP* may enhance the activity of cotton CSP protein to improve RNA chaperone activity, promote correct folding of RNA, stimulate gene transcription and translation, and ultimately augment protein activity and cotton cold resistance [47]. 

Previous studies have indicated that *AtCSPs* are highly expressed in the early stages of shoot tip and floral tissues and may play a role in various stages of plant development [18]. Accumulation of *CSPs* was observed in winter *T. aestivum* and *O. sativa* during the development of embryonic roots, hypocotyls, and crown tissues (nutrient growth stage), flowers, and seeds under low-temperature stress conditions [21,23]. These findings suggest that *CSPs* can impact plant growth and development. Some cis-acting elements of *CSP* genes in cotton are linked to growth and development. Additionally, *GhCSPs* have expression specificity in different tissues. For example, *GhCSP.A7* is highly expressed in floral organs, and *GhCSP.A1*, *GhCSP.A2*, and *GhCSP.A3* are expressed in rhizomes. From this, we hypothesized that the *CSPs* are involved in cotton growth and development.Integrated with the examination of the expression patterns of *GhCSP* at various sites and under diverse stresses, we determined that the genes specifically highly expressed in rhizomes and floral organs were aligned with those upregulated under low-temperature stress. The accumulation of CSP proteins at these sites may be observed during stress and nutrient growth stages.

In this study, we conducted genome-wide identification and expression pattern analysis of *CSP* gene family members from across four cotton species. According to our findings, we concluded that *CSP* genes may play important regulatory roles in cotton growth and development alongside low-temperature stress response. 

## 4. Materials and Methods

### 4.1. Identification of CSP Genes in Cotton

*G. hirsutum* and *G. barbadense* are allotetraploids, which evolved from two diploid cotton species, *G.arboreum* and *G.raimondii*. Therefore, these four cotton species were selected for gene family analysis. The whole genome sequence data and annotation information for four cotton species, *G. hirsutum* (HAU), *G. barbadense* (HAU), *G. arboreum* (CRI), and *G. raimondii* (JGI) (HAU, CRI, and JGI represent different assemblies of the genome), were obtained from the CottonFGD website (https://cottonfgd.net/ (accessed on 1 June 2023)) [48]. The *CSP* gene in *A. thaliana* was identified and downloaded from the TAIR database (https://www.arabidopsis.org/ (accessed on 3 June 2023)). A BUPP search was conducted in CottonFGD using the *A. thaliana* CSP protein sequence (*AT4g36020*, *AT4g38680*, *AT2g17870*, *AT2g21060*) with the e value set to 1e−10 to acquire the cotton CSP protein data. The Hidden Markov Models PF00313 (CSD) and PF00089 (zf-CCHC) corresponding to the *CSP* gene family were acquired from the PFAM database (http://pfam.xfam.org (accessed on 6 June 2023)) to characterize the *CSP* gene containing two domains in cotton [49]. The protein sequences from *A. thaliana* and cotton were aligned using HMM and Blastp, and domain information was identified using the CCD databases (http://www.NCBI.nlm.nih.gov/CDD/ (accessed on 9 June 2023)) of the SMART website (https://smart.embl.de/ (accessed on 9 June 2023)) and the NCBI website [50,51]. Following several screenings to remove ineligible proteins, the remaining non-redundant proteins were identified as cotton CSP proteins. The ExPAsy online tool (https://web.expasy.org/protparam/ (accessed on 15 June 2023)) was employed to determine the physicochemical parameters, including molecular weight, isoelectric point (pI), and protein length of CSP protein sequences across four cotton seeds [52]. WoLF PSORT tools (https://wolfpsort.hgc.jp/ (accessed on 21 June 2023)), CELLO (http://cello.life.nctu.edu.tw/ (accessed on 21 June 2023)), and YLoc (https://abi-services.cs.uni-tuebingen.de/yloc (accessed on 21 June 2023)) were utilized to predict subcellular localization of the proteins.

### 4.2. CSP Gene Subcellular Localization Analysis

The pCAMBIA1302 vector was digested with the restriction endonuclease *Nco*I to obtain a linear vector. Subsequently, the plasmids containing the target genes (*GhCSP.A1*, *GhCSP.A2*, *GhCSP.A3*) were ligated with the vector plasmid using a ClonExpress II One Step Cloning Kit (Vazyme) to create new vectors. The constructed vector was introduced into *Agrobacterium rhizogenes* GV3101 (pSoup) and then injected for transient expression on the underside of 4-week-old tobacco leaves. The infected area was subsequently excised for observation after 2 days of dark treatment. Bright-field fluorescence imaging of tobacco leaves was analyzed using a laser confocal super-resolution microscope (LSM980). All the primers used for this experiment are shown in Appendix A.

### 4.3. Phylogenetic Analysis

To investigate the evolutionary relationships between CSP proteins across different species, CSP protein sequences of *A. thaliana*, *O. sativa*, *T. aestivum*, *B. chinensis*, and *C. sinensis* were acquired from the Ensembl Plants database (https://plants.ensembl.org/index.html (accessed on 29 June 2023)) [53]. MEGA 11 was used for multiple sequence comparisons on the identified *CSP* sequences of different species, including cotton, and the phylogenetic tree was developed using the neighbor-joining method. The bootstrap value was established as 1000, and other parameters were set to default to obtain the NWK file. Finally, ITOL (https://itol.embl.de/ (accessed on 3 July 2020)) was applied to colorize this tree.

### 4.4. Conserved Motifs and Gene Structure Analysis of Cotton CSP Genes

TBtools software v2.031 was employed to trim and compare the protein sequences of cotton *CSPs*, and an evolutionary tree was developed using the IQtree function [54]. The online MEME tool (https://meme-suite.org/ (accessed on 12 July 2020)) was utilized to predict conserved motifs of CSP proteins. Employing the online tool GSDS (http://GSDS.cbi.pku.edu.cn/ (accessed on 22 July 2020)), we predicted the cotton CSPs’ genetic structure [55]. The NCBI conservative CDD structure domain database (https://www.ncbi.nlm.nih.gov/ (accessed on 26 July 2020)) was employed to predict the structure of the cotton *CSP* domain. Eventually, TBtools was employed to visualize the evolutionary tree, motif, domain, and gene structure.

### 4.5. Chromosomal Localization and Collinearity Analysis of Cotton CSP Genes

The chromosome distribution information of *CSP* genes was acquired from gene annotation files throughout the four cotton species (*G. arboreum*, *G. raimondii*, *G. barbadense*, and *G. hirsutum*) and visualized using TBtools. Repeat gene pair collinearity examination of the four cotton CSP protein sequences was conducted through the MCScanX program v2.031, and gene pairs meeting the criteria for tandem repeat genes were identified and visualized using TBtools. We calculated the ratio of the number of non-synonymous substitutions per non-synonymous site to the number of synonymous substitutions per synonymous site (Ka/Ks) for duplicated genes to investigate the selection pressure [55]. To examine the homology relationships of homologous *CSP* genes identified in cotton and other species, the multicollinearity scanning toolkit (MCScanX) was utilized to map the homology analysis [56].

### 4.6. Analysis of Cis-Acting Elements in CSP Genes from Cotton

The 2000 bp DNA sequence upstream of cotton *CSPs* was extracted based on the location of genes to analyze the promoter region of *CSPs* genes. Using PlantCare (http://bioinformatics.psb.ugent.be/webtools/PlantCare/html/ (accessed on 30 July 2020)) we identified the cis-acting regulatory elements in the promoter sequences, and used TBtools to visualize recognition elements.

### 4.7. RNA-Seq Analysis

The expression patterns of *GhCSP* genes across different parts and stresses were examined using original RNA-seq data. FPKM (fragments per kilobase of exon per million fragments mapped) indicated the amount of reads in each 1000 bases identified as a part of an exon across every million reads throughout the landscape. The FPKM approach was employed to examine gene expression. TBtools was used to construct a heatmap [35].

### 4.8. RT—qPCR Analysis

R15 (a conventional upland cotton cultivar) was planted in a blend of vermiculite and nutrient-rich soil, and cultivated in an incubator with a 16/8 h (light/dark) photoperiod and a temperature of 25 °C. Cotton seedlings aged three weeks were subjected to low-temperature stress (4 °C), while the control materials were left untreated. The leaves were collected at 0, 6, 12, and 24 h after treatment (3 repetitions), immediately frozen with liquid nitrogen, and stored at −80 °C. Total RNA was extracted from the samples using the FastPure Plant Total RNA Isolation Kit (RC401-01, Vazyme Biotech, Nanjing, China). The RNA was treated with DNase I to eliminate any DNA contamination. Reverse transcription was performed on 2 μg of total RNA using the HiScript III All-in-One RT SuperMix Perfect kit for qPCR, R333-01 (Vazyme). The ChamQ Universal SYBR qPCR Master Mix (Q711-02, Vazyme) and ABI 7500 Real-Time PCR system were employed for real-time quantitative PCR of the samples. Samples were analyzed using biological triplicates, and the internal reference gene was *GhUBQ7* and *GhHIS3*. The 2^−ΔΔCt^ method was employed to assess the relative expression levels of the target genes [57]. The data were analyzed for significant differences using one-way analysis of variance (ANOVA) and Student’s *t*-tests. Appendix A indicates the primer sequences.

## Figures and Tables

**Figure 1 plants-13-00643-f001:**
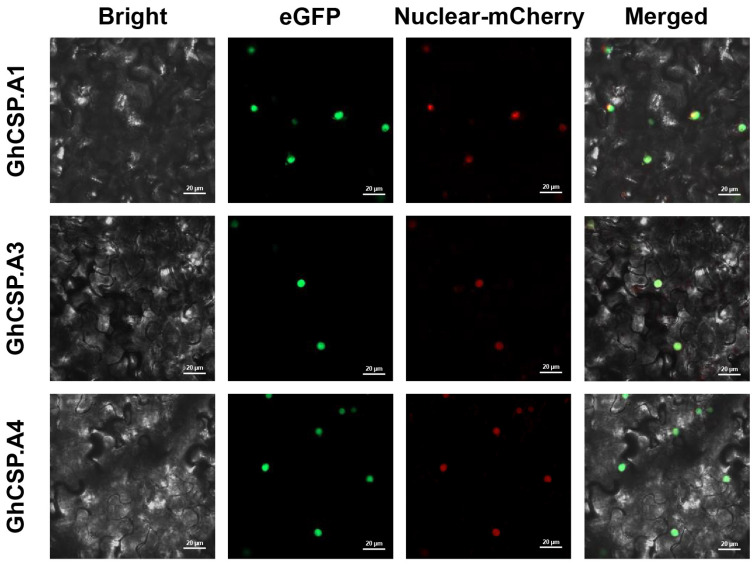
Subcellular localization of *GhCSP.A1-A3* proteins carried out in tobacco leaves. The Bright plot is the bright field. The green fluorescence in the eGFP plot represents the GFP fluorescence signal. The red fluorescence in the mCherry plot represents the organelle marker. The yellow fluorescence in the Merged plot represents the overlap of the green fluorescence from GFP with the red fluorescence from the marker.

**Figure 2 plants-13-00643-f002:**
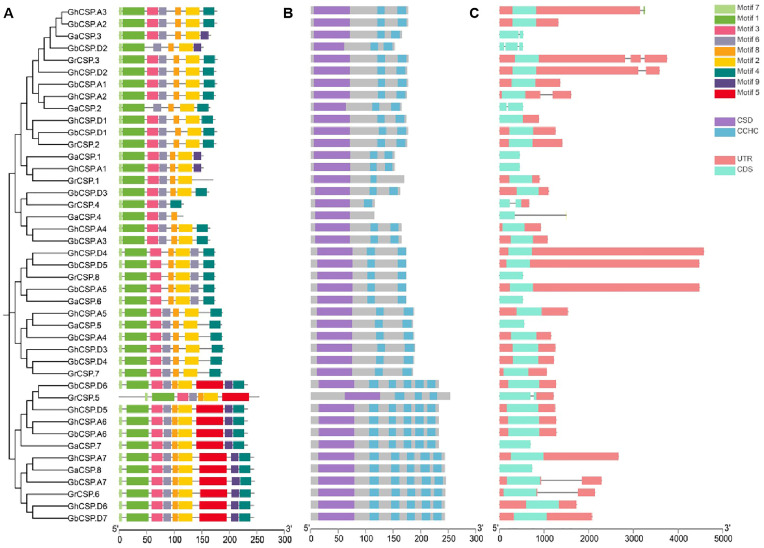
Identification of conserved structural domains as well as gene structure analysis of cotton *CSPs*. (**A**) Evolutionary tree and conserved motifs throughout the cotton *CSP* family. (**B**) Conserved structural domains of the cotton *CSP* family. (**C**) Exon-intron structures of the cotton *CSP* family. Squares indicate exons. Lines indicate introns.

**Figure 3 plants-13-00643-f003:**
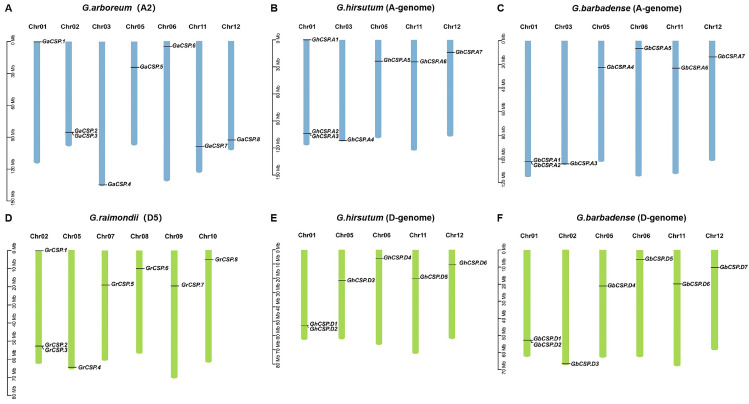
Chromosome distribution and duplication of *GhCSPs.* (**A**) Eight *GaCSPs* were mapped onto seven chromosomes in *G. arboreum*. (**B**) Seven *GhCSPs* were mapped onto five chromosomes in *G. barbadense* (A-genome). (**C**) Seven *GbCSPs* were mapped onto six chromosomes in *G. hirsutum* (A-genome). (**D**) Eight *GrCSPs* were mapped onto six chromosomes in *G. raimondii.* (**E**) Six *GhCSPs* were mapped onto five chromosomes in *G. barbadense* (D-genome). (**F**) Seven *GbCSPs* were mapped onto six chromosomes in *G. hirsutum* (D-genome).

**Figure 4 plants-13-00643-f004:**
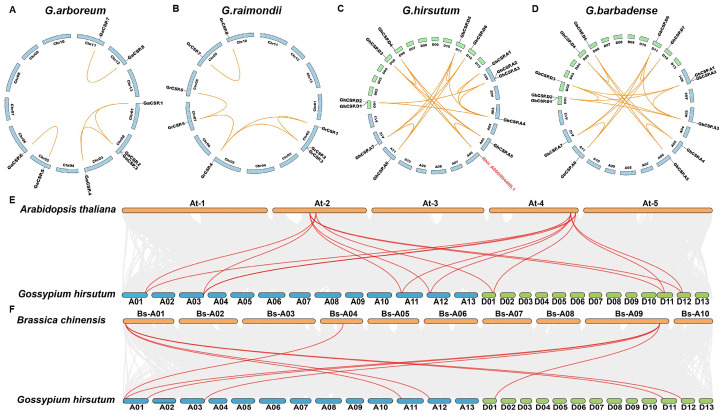
Synteny examination of the *GhCSP* genes. (**A**) *G. arboreum*; (**B**) *G. raimondii*; (**C**) *G. hirsutum*; (**D**) *G. barbadense*; (**E**) *A. thaliana* and *G. hirsutum*; (**F**) *B. chinensis* and *G. hirsutum*.

**Figure 5 plants-13-00643-f005:**
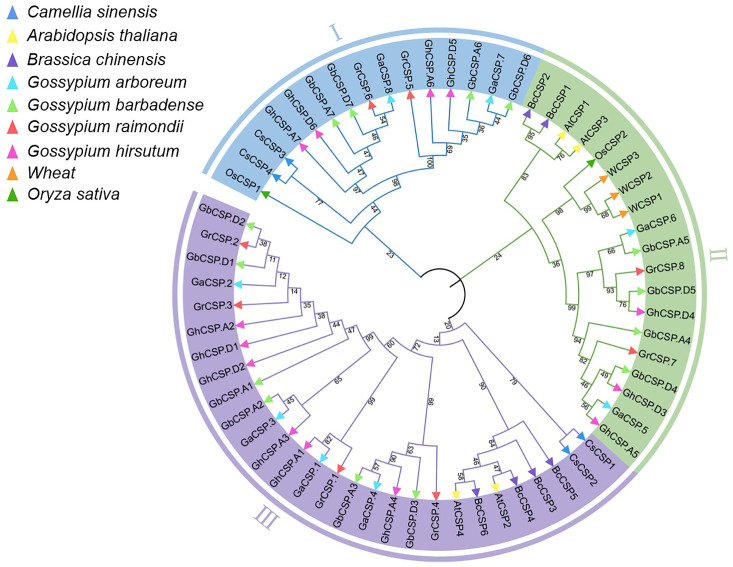
Phylogenetic examination of *CSP* gene families. Phylogenetic analysis of the *CSP* gene families across cotton, *A. thaliana*, *O. sativa*, *C. sinensis*, *B. chinensis*, and *T. aestivum*. The inner circle is marked in blue, green, and purple, representing the CSPI, CSPII, and CSPIII clades, respectively. Diverse plants are indicated by small triangles of different colors. The number alongside the branch indicates the bootstrap value of each group derived from 1000 replicates.

**Figure 6 plants-13-00643-f006:**
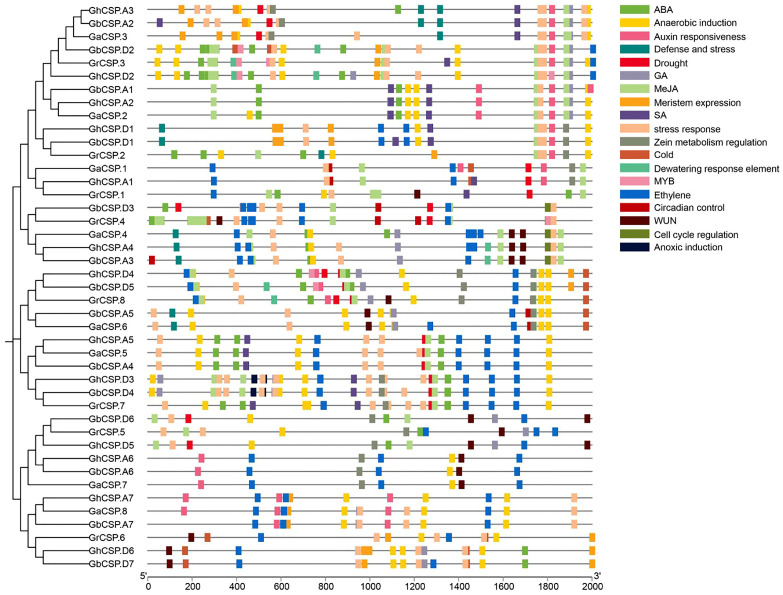
Predicted hormone response, stress response, and growth-related cis-elements within the promoter regions of *CSP* genes.

**Figure 7 plants-13-00643-f007:**
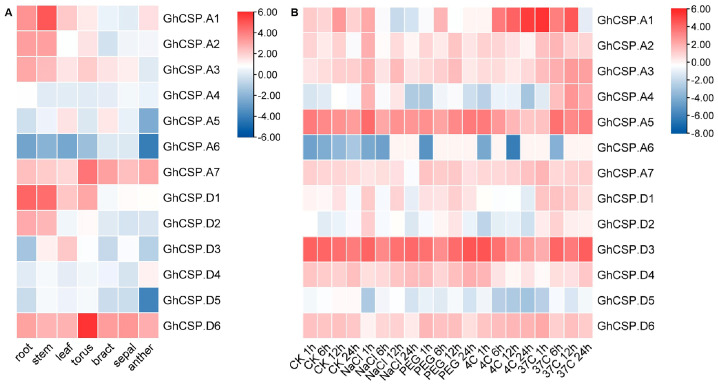
Expression pattern examination of *GhCSP.* (**A**) Analysis of the expression patterns of *GhCSP* across different organs. (**B**) Expression pattern analysis of *GhCSP* under control, low temperature, high temperature, salt, and drought stress conditions, expressed as CK−control, NaCl−salt (400 mM NaCl), PEG−drought (20% PEG), 4C−low temperature (4°), and 37C−high temperature (37°), respectively.

**Figure 8 plants-13-00643-f008:**
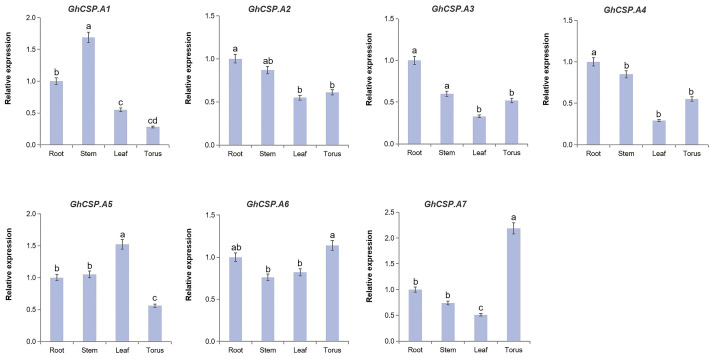
The expression of *GhCSP.A1–A7* across different organs. The qRT-PCR result of *GhCSP.A1–A7* in the roots, stems, leaves, and flowers of *G. hirsutum*. Values are presented as the mean ± S.D (*n* = 3 replicates). Significant differences were assessed using one-way analysis of variance (ANOVA) with *p* < 0.05.

**Figure 9 plants-13-00643-f009:**
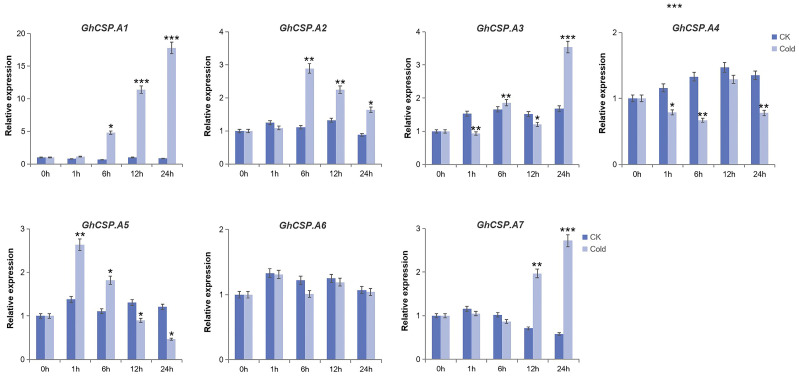
Expression patterns of seven *GhCSP* genes responding to cryogenic treatment. Each sample was sampled at 0, 1, 6, 12, and 24 h following treatment. CK was the control group. Cold was the treatment group. The findings are presented as the mean of three replicates. Significant differences are indicated by * *p* < 0.05, ** *p* < 0.01, and *** *p* < 0.001, determined using a Student’s *t*-test.

## Data Availability

For privacy reasons, data can be provided via private mail.

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
