# Peer review of "Genome-Wide Identification and Expression Analysis Unveil the Involvement of the Cold Shock Protein (CSP) Gene Family in Cotton Hypothermia Stress"

_plants, 2024, doi:10.3390/plants13050643_

Round 1
Reviewer 1 Report
Comments and Suggestions for Authors
This manuscript entitled “Genome-Wide Identification and Expression Analysis Unveil the Involvement of the Cold Shock Protein (CSP) Gene Family in Cotton Under Hypothermia Stress” showcased the characterization cotton CSP family mainly through bioinformation analysis. The subcellular location and qRT-PCR analyses were employed to solid the results. The data is meanings for further researching plant CSP family.
Some concerns:
1. title needs to be revised. it appears to delete the "under".
2. In Abstract, Latin name words replaces Wheat.
3. Throughout the manuscript, some latin names need to be written by italic fond, e.g. Escherichia coli, Triticum aestivum.
4. Line 78, “Overall, the CSP family of proteins”, delete “of”.
5. Line 86, “spring” changed into “Spring”
6. In cell distribution of CSPs, the only mito location CSP needs to be verified.
7. Figure 2C, label the graphic explanation of exon and intron.
8. Line 248-250, this sentence meaning is unclear.
9. In “p < 0.05, ** p < 0.01, ***p < 343 0.001and ****p < 0.0001, determined using a Student’s t-tes” the “p” and “t” should be written in italic fond.
Comments on the Quality of English Languageminor revision
Author Response
Response to Reviewer 1 Comments
We are thankful for Reviewer 1 meaningful comments that improved our manuscript greatly.
Point 1: title needs to be revised. it appears to delete the "under".
Response 1: Changed.
Point 2: In Abstract, Latin name words replaces Wheat.
Response 2: Changed.
Point 3: Throughout the manuscript, some latin names need to be written by italic fond, e.g. Escherichia coli, Triticum aestivum.
Response 3: Changed.
Point 4: Line 78, “Overall, the CSP family of proteins”, delete “of”.
Response 4: Changed.
Point 5: Line 86, “spring” changed into “Spring”
Response 5: Changed.
Point 6: In cell distribution of CSPs, the only mito location CSP needs to be verified.
Response 6: Because the amplification of the GbCSP.D2 gene took a long time without success, the three fastest amplified genes were chosen.
Point 7: Figure 2C, label the graphic explanation of exon and intron.
Response 7: Added.
Point 8: Line 248-250, this sentence meaning is unclear.
Response 8: We have made new instructions.
Point 9:In “p < 0.05, ** p < 0.01, ***p < 343 0.001and ****p < 0.0001, determined using a Student’s t-tes” the “p” and “t” should be written in italic fond.
Response 9: Changed.
Reviewer 2 Report
Comments and Suggestions for Authors
This work focused on the characterisation of the Cold Shock Protein (CSP) gene family in cotton under hypothermia stress. This study is fundamental and interesting because the data obtained during the study can be used to develop technologies to increase cotton production under cold conditions. In general, the authors have carried out an interesting research, but I have some suggestions regarding the manuscript.
General remarks:
1. The introduction is too general. What are the general mechanisms of plant resistance to low temperature? What are the main proteins that confer cold stress resistance in plants? How many genes are represented by the multigenic cold shock protein (CSP) family in different plants? What triggers the expression of CSP genes (possibly transcription factors)? What is their protein function? What are their target proteins? The data given in lines 88-91 are fragmented, as the authors have not previously indicated the main mechanisms of plant adaptation to cold stress. It would not be superfluous to indicate the characteristics of the cotton genome, what kind of ploidy it may have. The authors need to substantially revise the introduction.
2. Results. Table 1 is unreadable, it should be enlarged.
Why were the GhCSP A1/A2/A3 genes chosen for the analysis of cellular localisation?
Chapter 2.3. In this chapter, the authors started to discuss their results, referring to other work. It may be better to move this part to the Discussion chapter and leave only the results in this chapter.
In Figure 2, green and yellow indicate several values: green Motif 1, CSD and UTR, yellow zf-CCHC and CDS. The authors need to change the colours so that the reader is not confused as to what applies to what.
Chapter 2.4. Suggestions "It was found that 13 GhCSP genes were located on 10 chromosomes, of which seven were located on five chromosomes of the At genome and the remaining six were located on five chromosomes of the Dt chromosome group, according to the annotation information in the cotton genome (Figure 3)" and " The GaCSP gene was localized on seven chromosomes, distributed across chromosomes 01 (GaCSP.1), 02 (GaCSP.2 and GaCSP.3), 03 (GaCSP.4), 05 (GaCSP.5), 06 (GaCSP.6), 11 (GaCSP.7), and 12 (GaCSP.8).” contradict each other. Perhaps the text should be rewritten to make it easier to understand.
Figure 4a is illegible, it should be enlarged.
Lines 248-249. The sentence needs to be rewritten. The second part of the sentence partially duplicates the first. It would also be useful to give the Latin names of the plants whose genes were used for the phylogenetic analysis.
Figure 7. The effects should be deciphered in the set. CK - control, NaCl - salt, what is the concentration? PEG, etc.
3. Discussion. This section should also be rewritten. To draw more attention to the data obtained. There may be information on the involvement of CSP genes in the growth processes of other plants.
Lines 429-431 should be deleted. These are the results of your previous work, it is strange to put them in the last paragraph. Perhaps this should be mentioned in the body of the discussion.
4. Materials and methods. Lines 436-437. Why did the authors choose these cotton varieties? What is the reason for this choice?
What do the abbreviations in brackets mean: G. hirsutum (HAU), G. barbadense (HAU), G. arboreum (CRI) and G. raimondii (JGI)?
Line 456. What kind of genetic constructs have been obtained, what target genes have been used? Which plasmids were used to obtain binary constructs? Which restriction enzymes were used? How exactly were the agrobacteria inoculated onto a tobacco leaf? At what time was the localisation of the CSP proteins analysed? I also do not have a supplementary file that specifies the primers used to obtain genetic constructs.
Lines 503-505. Where exactly was the cotton grown: greenhouse, climate chamber, open field? How long were the cotton plants exposed to cold stress? Was the control group not stressed? This information is available in the results but needs to be added to this section.
What method was used to isolate RNA? What set of reagents was used to produce cDNA? Why was only one UBQ reference gene used? Usually at least 2, preferably 4 internal controls should be used.
- Minor Revisions
1) In the text. The Latin name of the species (plants and bacteria) is written in full only once, then the species name is abbreviated. The Latin names of plants and bacteria should be written in italics (also in figures). There is no full Latin name for cotton in the abstract and text. For example, Arabidopsis thaliana => A. thaliana.
2) In the text: p-value (p < 0.01) should be italicised;
Decision: - Accept after major essential revisions.
Comments on the Quality of English LanguageEnglish editing is required. Difficult to understand in places.
Author Response
Response to Reviewer 2 Comments
We are thankful for Reviewer 2 meaningful comments that improved our manuscript greatly.
Point 1: The introduction is too general. What are the general mechanisms of plant resistance to low temperature? What are the main proteins that confer cold stress resistance in plants? How many genes are represented by the multigenic cold shock protein (CSP) family in different plants? What triggers the expression of CSP genes (possibly transcription factors)? What is their protein function? What are their target proteins? The data given in lines 88-91 are fragmented, as the authors have not previously indicated the main mechanisms of plant adaptation to cold stress. It would not be superfluous to indicate the characteristics of the cotton genome, what kind of ploidy it may have. The authors need to substantially revise the introduction.
Response 1: Good suggestion!We have improved the introduction.
Point 2: Results. Table 1 is unreadable, it should be enlarged.
Why were the GhCSP A1/A2/A3 genes chosen for the analysis of cellular localisation?
Response 2: Inserted in the supplementary document. We randomly selected genes for validation.
Point 3: Chapter 2.3. In this chapter, the authors started to discuss their results, referring to other work. It may be better to move this part to the Discussion chapter and leave only the results in this chapter.
Response 3: We moved the sentence to the discussion section.
Point 4: In Figure 2, green and yellow indicate several values: green Motif 1, CSD and UTR, yellow zf-CCHC and CDS. The authors need to change the colours so that the reader is not confused as to what applies to what.
Response 4: Changed.
Point 5: Chapter 2.4. Suggestions "It was found that 13 GhCSP genes were located on 10 chromosomes, of which seven were located on five chromosomes of the At genome and the remaining six were located on five chromosomes of the Dt chromosome group, according to the annotation information in the cotton genome (Figure 3)" and " The GaCSP gene was localized on seven chromosomes, distributed across chromosomes 01 (GaCSP.1), 02 (GaCSP.2 and GaCSP.3), 03 (GaCSP.4), 05 (GaCSP.5), 06 (GaCSP.6), 11 (GaCSP.7), and 12 (GaCSP.8).” contradict each other. Perhaps the text should be rewritten to make it easier to understand.
Response 5: Here one is for land cotton and one for Asian cotton, two different cotton varieties.
Point 6: Figure 4a is illegible, it should be enlarged.
Response 6: Changed.
Point 7: Lines 248-249. The sentence needs to be rewritten. The second part of the sentence partially duplicates the first. It would also be useful to give the Latin names of the plants whose genes were used for the phylogenetic analysis.
Response 7: Changed.
Point 8: Figure 7. The effects should be deciphered in the set. CK - control, NaCl - salt, what is the concentration? PEG, etc.
Response 8: Added
Point 9:Discussion. This section should also be rewritten. To draw more attention to the data obtained. There may be information on the involvement of CSP genes in the growth processes of other plants.
Response 9: Changed. We have improved our discussion section.
Point 10:Lines 429-431 should be deleted. These are the results of your previous work, it is strange to put them in the last paragraph. Perhaps this should be mentioned in the body of the discussion.
Response 10: Changed. We moved the sentence to the discussion section.
Point 11:Materials and methods. Lines 436-437. Why did the authors choose these cotton varieties? What is the reason for this choice?
What do the abbreviations in brackets mean: G. hirsutum (HAU), G. barbadense (HAU), G. arboreum (CRI) and G. raimondii (JGI)?
Response 11: Upland cotton (G.hirsutum) and sea island cotton (G.barbadense) are allotetraploids, which evolved from two diploid cotton, Asian cotton (G.arboreum) and Raymond cotton (G. raimondii), so these four cotton species were selected for gene family analysis. The abbreviations in parentheses refer to different assemblies of the genome.
Point 12:Line 456. What kind of genetic constructs have been obtained, what target genes have been used? Which plasmids were used to obtain binary constructs? Which restriction enzymes were used? How exactly were the agrobacteria inoculated onto a tobacco leaf? At what time was the localisation of the CSP proteins analysed? I also do not have a supplementary file that specifies the primers used to obtain genetic constructs.
Response 12: Added
Point 13:Lines 503-505. Where exactly was the cotton grown: greenhouse, climate chamber, open field? How long were the cotton plants exposed to cold stress? Was the control group not stressed? This information is available in the results but needs to be added to this section.
What method was used to isolate RNA? What set of reagents was used to produce cDNA? Why was only one UBQ reference gene used? Usually at least 2, preferably 4 internal controls should be used.
Response 13: Added
Point 14:1) In the text. The Latin name of the species (plants and bacteria) is written in full only once, then the species name is abbreviated. The Latin names of plants and bacteria should be written in italics (also in figures). There is no full Latin name for cotton in the abstract and text. For example, Arabidopsis thaliana => A. thaliana.
2) In the text: p-value (p < 0.01) should be italicised;
Response 14: Changed.
Reviewer 3 Report
Comments and Suggestions for Authors
This paper describes the identification and characterization of Cotton genes involved in Cold-stress, particularly Cold-Shock Protein family members using both bioinformatic and wet-bench approaches. The authors identified 62 different genes in this category across 4 species of cotton. They also identified protein domains, cis-elements and did phylogenetic comparisons in their study to visualize evolutionary relationships in this gene family as well as hypothesize about selective pressures after gene duplication events. Wet-bench assays were used for sub-cellular localization of CSP in a tobacco system as well as RT-PCR.
Overall, the paper is well-written and the results are not overstated in the discussion. However, there are a few issues that need to be addressed before this paper is publication-ready. My comments are noted below.
Line 13 Add "and wet-bench"
LInes 50-53 seem irrelevant
Line 55 CCHC and CSD. Spell out all acronyms first or describe them.
Line 63 and throughout manuscript: The way the plant names are written is inconsistent. In some places they are lowercase, in others it is upper case. In some cases they are italicized. Scientific and common names are used interchangeably. My comment to this is be consistent. Scientific names need to be written out completely as Genus species the first time and then can be abbreviated (ex. line 114, 408) or the common name can be used.
LIne 94 needs a citation.
Line 114 is unclear. How does this indicate that this gene family is conserved. Please explain in your manuscript.
LIne 165-166 You need 1-2 sentences and a reference stating why genes might not have introns. What is the evolutionary importance of this?
LIne 98 Has not have
Line 138. Table 1 can be supplementary. I am not sure it shows anything significant since all your genes located to the nucleus (except 1). This can just be stated in the results.
Line 192 What does Dt stand for?
Figure 3 and Figure 4- the labels in figure 3 and 4 are too small to read. Please enlarge.
Lines 218-227 Admittedly, I have never used Ka/Ks values. Please cite another reference in your paper where this method has been done before.
Figure 6- You used the Plant CARE data base to find motifs, but I think you need to do a random set of genes as well. Are the cis-elements on these genes unique to this family or like most cis-elements are they found in a lot of other genes in the genome? I think you need to look to see whether these cis elements are uniquely enriched in this family vs. which elements are just present. Otherwise, it does not tell you much about how they might be transcriptionally controlled.
Line 400 You say "studies" but then only cite one reference. Please cite additional papers.
Figures 8 & 9- the most egregious error in this paper is the statistical methods used to analyze the RT-PCR data. Using a student's T-Test to compare more than one treatment (independent variable) is in appropriate. An ANOVA (analysis of variance should be done if comparing across multiple tissues and times). This may affect the results and you may need to change your findings based on the new statistics. These should also be described clearly in the methods section.
Line 456 The methods for creating vectors and transfecting tobacco cells is inadequately described. The readers should be able to replicate your study and "vectors were constructed and introduced" is not enough.
Comments on the Quality of English Language
English is fine. Minor errors.
Author Response
Response to Reviewer 3 Comments
We are thankful for Reviewer 3 meaningful comments that improved our manuscript greatly.
Point 1: Line 13 Add "and wet-bench"
Response 1: I'm sorry, we don't quite understand where to add it?
Point 2: LInes 50-53 seem irrelevant
Response 2:Deleted.
Point 3: Line 55 CCHC and CSD. Spell out all acronyms first or describe them.
Response 3: Changed.
Point 4: Line 63 and throughout manuscript: The way the plant names are written is inconsistent. In some places they are lowercase, in others it is upper case. In some cases they are italicized. Scientific and common names are used interchangeably. My comment to this is be consistent. Scientific names need to be written out completely as Genus species the first time and then can be abbreviated (ex. line 114, 408) or the common name can be used.
Response 4: Changed.
Point 5: Line 94 needs a citation.
Response 5: Added.
Point 6: Line 114 is unclear. How does this indicate that this gene family is conserved. Please explain in your manuscript.
Response 6: We have inserted.
Point 7: LIne 165-166 You need 1-2 sentences and a reference stating why genes might not have introns. What is the evolutionary importance of this?
Response 7: Good suggestion!We have inserted in the discussion section.
Point 8: LIne 98 Has not have
Response 8: Changed.
Point 9:Line 138. Table 1 can be supplementary. I am not sure it shows anything significant since all your genes located to the nucleus (except 1). This can just be stated in the results.
Response 9: Changed. Inserted in the supplementary document.
Point 10:Line 192 What does Dt stand for?
Response 10: Upland cotton (G.hirsutum) and sea island cotton (G.barbadense) are tetraploid cotton, and the genome of tetraploid cotton includes the A genome and the D genome, which are usually represented by At and Dt.
Point 11: Figure 3 and Figure 4- the labels in figure 3 and 4 are too small to read. Please enlarge.
Response 11: We've enlarged the labels as much as possible.
Point 12: Lines 218-227 Admittedly, I have never used Ka/Ks values. Please cite another reference in your paper where this method has been done before.
Response 12: Added.
Point 13:Figure 6- You used the Plant CARE data base to find motifs, but I think you need to do a random set of genes as well. Are the cis-elements on these genes unique to this family or like most cis-elements are they found in a lot of other genes in the genome? I think you need to look to see whether these cis elements are uniquely enriched in this family vs. which elements are just present. Otherwise, it does not tell you much about how they might be transcriptionally controlled.
Response 13: This part is about analyzing what functions these genes might have by predicting what cis-elements they have, rather than looking at how they are transcribed.
Point 14:Line 400 You say "studies" but then only cite one reference. Please cite additional papers.
Response 14: Added
Point 15:Figures 8 & 9- the most egregious error in this paper is the statistical methods used to analyze the RT-PCR data. Using a student's T-Test to compare more than one treatment (independent variable) is in appropriate. An ANOVA (analysis of variance should be done if comparing across multiple tissues and times). This may affect the results and you may need to change your findings based on the new statistics. These should also be described clearly in the methods section.
Response 15: Changes have been made to Figure 8. Figure 9 shows comparisons between the control and treatment groups at different times, so the Student's t-test was used.
Point 16:Line 456 The methods for creating vectors and transfecting tobacco cells is inadequately described. The readers should be able to replicate your study and "vectors were constructed and introduced" is not enough.
Response 16: Added
Round 2
Reviewer 2 Report
Comments and Suggestions for Authors
Authors significantly improved the manuscript. However, before this manuscript can be published, some minor improvements should be done:
1. Figure 2 «Squares indicate exons. Lines indicate endosperm.» Why endosperm? Maybe introns?
2. Why do you write the name of the primers in the Supplementary Table 2: actin-GhUBQ7-F, actin-GhUBQ7-R, actin-GhHIS3-F, actin-GhHIS3-R? Why actin? Perhaps you should remove this word from the name of the primer or change on «internal reference gene».
3. «Response 11: Upland cotton (G.hirsutum) and sea island cotton (G.barbadense) are allotetraploids, which evolved from two diploid cotton, Asian cotton (G.arboreum) and Raymond cotton (G. raimondii), so these four cotton species were selected for gene family analysis. The abbreviations in parentheses refer to different assemblies of the genome.» Perhaps this sentence should be added to the text of the manuscript.
4. «Subsequently, the plasmids containing the target genes (GhCSP.A1, 452 GhCSP.A2, GhCSP.A3) were ligated with the vector plasmid using a seamless cloning 453 method to generate new vectors.» A reference to the method is required.
5. Agrobacterium rhizogenes => Agrobacterium rhizogenes
6. Student's t-tests => Student's t-tests
Decision: - Accept after minor essential revisions (which the authors can be trusted to make)
Comments on the Quality of English LanguageEnglish language fine. Maybe minor editing of English language required
Author Response
Response to Reviewer 2 Comments
We are thankful for Reviewer 2 meaningful comments that improved our manuscript greatly.
Point 1: Figure 2 «Squares indicate exons. Lines indicate endosperm.» Why endosperm? Maybe introns?
Response 1: Changed,Sorry for the mistake.
Point 2: Why do you write the name of the primers in the Supplementary Table 2: actin-GhUBQ7-F, actin-GhUBQ7-R, actin-GhHIS3-F, actin-GhHIS3-R? Why actin? Perhaps you should remove this word from the name of the primer or change on «internal reference gene».
Response 2: Changed,Sorry for the mistake.
Point 3: «Response 11: Upland cotton (G.hirsutum) and sea island cotton (G.barbadense) are allotetraploids, which evolved from two diploid cotton, Asian cotton (G.arboreum) and Raymond cotton (G. raimondii), so these four cotton species were selected for gene family analysis. The abbreviations in parentheses refer to different assemblies of the genome.» Perhaps this sentence should be added to the text of the manuscript.
Response 3: Added.
Point 4: «Subsequently, the plasmids containing the target genes (GhCSP.A1, 452 GhCSP.A2, GhCSP.A3) were ligated with the vector plasmid using a seamless cloning 453 method to generate new vectors.» A reference to the method is required.
Response 4: Here we have used kits and made changes to the text.
Point 5: Agrobacterium rhizogenes => Agrobacterium rhizogenes
Response 5: Changed.
Point 6: Student's t-tests => Student's t-tests
Response 6: Changed.
